# DISCO: A Large Scale Human Annotated Corpus for Disfluency Correction in Indo-European Languages

**Vineet Bhat, Preethi Jyothi, Pushpak Bhattacharyya**
Indian Institute of Technology Bombay, India

## Abstract

Disfluency correction (DC) is the process of removing disfluent elements like fillers, repetitions and corrections from spoken utterances to create readable and interpretable text. DC is a vital post-processing step applied to Automatic Speech Recognition (ASR) outputs, before subsequent processing by downstream language understanding tasks. Existing DC research has primarily focused on English due to the unavailability of large-scale open-source datasets. Towards the goal of multilingual disfluency correction, we present a high-quality human-annotated DC corpus covering four important Indo-European languages: English, Hindi, German and French. We provide extensive analysis of results of state-of-the-art DC models across all four languages obtaining F1 scores of 97.55 (English), 94.29 (Hindi), 95.89 (German) and 92.97 (French). To demonstrate the benefits of DC on downstream tasks, we show that DC leads to 5.65 points increase in BLEU scores on average when used in conjunction with a state-of-the-art Machine Translation (MT) system. We release code to run our experiments along with our annotated dataset here[1].

## 1 Introduction

Humans often think and speak simultaneously in conversations, introducing erroneous words in utterances (Gupta et al., 2021). These words do not contribute to semantics of a sentence and hence can be removed to create fluent and easy-to-interpret utterances. Disfluency Correction (DC) is defined as the removal of such disfluent elements from spoken utterances (Shriberg, 1994).

**Motivation:** Apart from making sentences readable and interpretable, DC also helps downstream natural language processing tasks like Machine Translation (MT) (Rao et al., 2007; Wang et al., 2010). Removing disfluencies shortens sentences, making it easier for automatic MT systems to translate these utterances. Moreover, the removed erroneous words are not translated which makes the output translation fluent containing all semantics from the source sentence. Table 1 illustrates examples where Google MT produces disfluent and difficult-to-read English translations of disfluent sentences in 3 languages - Hindi, German and French, establishing the need for DC.

| Disfluent Sentence | Google MT output |
|---|---|
| वाच में रनिंग अह्ह्ह स्मार्ट वॉच में रोका हुआ रनिंग टाइमर रिज्यूम करो | running in watch ahhh resume running paused timer in smart watch |
| je veux je veux euh enregistrer une une euh vidéo sur instagram | I want I want uh record a uh video on instagram |
| ich brauche eine fahrt äh eine fahrt zum bahnhof in einer stunde | I need a ride er a ride to the train station in an hour |

Table 1: English translations produced by Google MT for disfluent sentences in Hindi, French and German. All disfluent words are marked in red.

Previous work in DC has leveraged variety of machine learning models for removing disfluent utterances from text (Ostendorf and Hahn, 2013b; Rasooli and Tetreault, 2015; Zayats et al., 2016b). However, data in DC is scarce, limiting the use of large transformer models. Switchboard (Godfrey et al., 1992), the most extensively available open-source DC corpus, contains English spoken utterances with only 5.9% disfluent words in the entire dataset (Charniak and Johnson, 2001). Synthetic Data Generation (SDG) has emerged as a viable solution to the data scarcity problem (Passali et al., 2022; Kundu et al., 2022). However, SDG can be challenging as it needs expert grammatical knowl-

---

[1] https://github.com/vineet2104/DISCO

| Disfluency Type | Description | Example |
|---|---|---|
| Filler | Words like *uhh*, *err*, *uhmm* that are often uttered to retain turn of speaking. Each language has a different set of filler words commonly uttered. | EN: Write a message to um Sarah.
DE: Fortsetzen ähm meines Lauftrainings.
FR: Montre euh mes applications.
HI: मेरा उम्म पल्सरेट फिटबिट में चेक करो |
| Repetition | Consists of words or phrases that are repeated in conversational speech | EN: Add this number to my to my contacts.
DE: ein Instagram-Foto machen machen.
FR: Enregistre mes 400 calories enregistre.
HI: क्या तुम हॉस्पिटल का हॉस्पिटल का एक नोट बना सकते हो? |
| Correction | Disfluencies that consist of words incorrectly spoken and immediately corrected with a fluent phrase | EN: Get me the order my order status on the desk chair I ordered from Overstock.
DE: HD Video auf aufnehmen.
FR: Reprendre l'exercice d'étirem d'étirement
HI: रा राहु राहुल का मैसेज पढ़ो |
| False Start | Examples where the speaker changes their chain-of-thought mid sentence to utter a completely different fluent phrase | EN: In an email let's email Tom Hardy about Saturday's video shoot.
DE: Facebook uh Jahr Facebook bitte.
FR: Envoi de le envoi du SMS à maman.
HI: कल उम्म आज की ब्लड प्रेशर रीडिंग बताओ |
| Fluent | Examples which do not contain any disfluent words or phrases | EN: Can you make a note for Johnny that says dinner at eight on my laptop?
DE: Nummer zu Kontakten hinzufügen..
FR: Je veux j'aimerais ouvrir TikTok..
HI: क्या आप योसेमाइट नेशनल पार्क को ईमेल कर सकते हैं? |

Table 2: Types of sentences observed in the DISCO corpus. All disfluencies are marked in red; EN-English, DE-German, FR-French, HI-Hindi. Examples in languages other than English, with their corresponding gloss and transliteration can be found in Appendix E

edge and the data created can often fail to mimic complex disfluencies encountered in real-life dialogues (Gupta et al., 2021).

Hence there is a dire need to develop DC datasets with utterances from real-life conversational situations. Existing datasets have focused on increasing the available data in English. This paper presents a high-quality DC corpus in English and widely spoken languages like Hindi, German and French. Our dataset significantly expands the available data in English and Hindi. To the best of our knowledge, we are the first to create an open-source DC corpus for German and French[2]. Our contributions are:

1. A human-labeled dataset of 12K+ disfluent-fluent text utterance pairs in 4 languages: English, Hindi, German and French with extensive data analysis (Section 3.4).

2. Experimenting with various state-of-the-art techniques for DC ranging from traditional ML models to large transformers (Table 5). Our best models (fine-tuned multilingual transformers) achieve an F1 score of 97.55 (English), 94.29 (Hindi), 95.89 (German) and 92.97 (French). Our results in English and Hindi are competitive with other approaches, but we do not report direct improvement due to the different testing datasets used.

3. Improving BLEU score of a state-of-the-art MT system by 5.65 points in Hindi-English and German-English language pairs after automatic disfluency removal from source sen-

---

[2]Although Cho et al. (2014) annotated the KIT lecture corpus (Stüker et al., 2012) for disfluencies in German, their data is not shared publically.

tences (Table 10). Similar analyses for other language pairs are a part of our future work.

## 2 Related Work

The study of disfluencies as a spoken language phenomenon was first proposed in Shriberg (1994). DC has been established as a vital post-processing task for ASR transcripts (Rao et al., 2007; Wang et al., 2010). Although earlier DC systems were based on translation methods (Honal and Schultz, 2003), current research covers two additional methods: parsing-based and sequence tagging-based techniques. Translation-based methods use a noisy channel approach towards DC hypothesizing that disfluent sentences are fluent sentences with noisy elements (Jamshid Lou and Johnson, 2017; Johnson and Charniak, 2004; Zwarts and Johnson, 2011). Parsing-based methods use techniques such as dependency parsing to predict syntactic structure of an utterance along with disfluent elements (Rasooli and Tetreault, 2015; Honnibal and Johnson, 2014; Wu et al., 2015). Sequence tagging methods work well for disfluency removal from real-life spoken utterances, assigning disfluent/fluent label to every word in the sentence (Hough and Schlangen, 2015a; Ostendorf and Hahn, 2013a; Zayats et al., 2016a; Chen et al., 2022). Language clues and part-of-speech tags based systems have also been explored for DC (Bove, 2008; Christodoulides et al., 2014). There is a notable gap in literature regarding real data annotation in DC, with Switchboard (Godfrey et al., 1992) and Salesky et al. (2018) being the most extensive open-source labeled datasets for English DC. Although Gupta et al. (2021) introduced a dataset for disfluencies in English question answering, they have not been annotated for disfluent words. Without labeled data, various zero-shot, few-shot, and multi-task learning techniques have been proposed, which train on multilingual data, creating and utilizing synthetically generated disfluent sentences (Wang et al., 2018; Passali et al., 2022; Kundu et al., 2022; Bhat et al., 2023). In this work, we experiment with sequence tagging methods for DC.

## 3 DISCO: A Dataset for Disfluency Correction

This section analyzes the DISCO corpus, created with the help of English, Hindi, German and French language experts. DISCO contains paral-lel disfluent-fluent sentence pairs in the above four languages and English translations of fluent sentences in Hindi and German along with disfluency and domain labels.

### 3.1 Terminology

Shriberg (1994) defines disfluencies as a composition of Reparandum, Interregnum and Repair (Figure 1). *Reparandum* refers to words erroneously uttered by the speaker. The speaker acknowledges that a previous utterance might be incorrect using *interregnum*, whereas *repair* contains words that correct mis-spoken words. Disfluent utterances might consist of an interruption point- a spoken phenomena like speech pauses. DC removes reparandum and interregnum while retaining repair to make the output sentence more fluent.

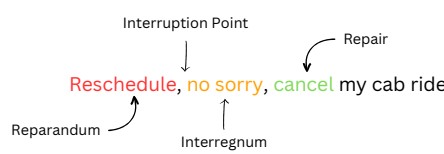

Figure 1: A disfluent utterance in English, marked with various components of disfluencies.

We study four types of disfluencies observed in our dataset: Filler, Repetition, Correction and False Start. Additionally, there are some fluent sentences present in our corpus. Table 2 describes each type of sentence with some real examples from the DISCO dataset.

### 3.2 Data Collection Method

Goel et al. (2023) released an open-source dataset containing real-life utterances of humans with AI agents for task-oriented dialogue parsing. We extract disfluent sentences and domain labels in English, Hindi, German and French from this corpus. These utterances consist of human dialogues like making notes, monitoring fitness, adding new contacts, opening apps, etc. All sentences are shared with respective language experts for fluent sentence creation and disfluency-type annotation.

### 3.3 Annotation Protocol and Challenges

For each language, we hired external annotators from reputed translation agencies with experience in data annotation. They were asked to create fluent sentences corresponding to disfluent utterances

along with disfluency type labels. Each annotator was paid competitively based on current market standards (approximately $ 0.018 per word). Since we follow a sequence tagging approach towards DC, the annotators were asked to only remove disfluent words from utterances without changing word order or correcting original words/phrases.

Due to budget constraints, we could not utilize the entire dataset in German and French from Goel et al. (2023). However, we carefully select sentences in these languages to sufficiently cover all disfluency types with varied length and complexity of utterances. Table 3 summarizes the total amount of data created and the amount of disfluency present in the corpus.

| Lang | No. of sentence pairs | No. of words | % disfluent words |
|------|------|------|------|
| En | 3479 | 31994 | 18.99 |
| Hi | 3180 | 32435 | 18.99 |
| De | 3096 | 22451 | 20.93 |
| Fr | 3005 | 22489 | 17.72 |

Table 3: Total count of disfluent-fluent pairs in DISCO and percentage of disfluency present; En-English, Hi-Hindi, De-German, Fr-French.

Since for every language, only one annotator created fluent sentences and disfluency type labels, ensuring high quality data was very important. We strongly encouraged the annotators to flag all dubious instances, after which the authors take a majority vote of retaining/removing doubtful disfluent words using high quality translation tools and subject knowledge wherever necessary. Flagged examples and our reasoning for specific annotations have been discussed in Appendix A.

### 3.4 Key Statistics

The DISCO corpus is carefully created to ensure healthy representation of various disfluency types and complexity of sentences. Table 4 describes average length of disfluent and fluent sentences for each language. Our analysis shows that in similar context, commands to AI agents are shorter in German and French than in English and Hindi. The standard deviation of the disfluent sentences demonstrates that the dataset also contains longer utterances, more than ten words long, in each language that are relatively difficult to correct. We showcase the distribution of disfluent sentences across disfluency types in figure 2.

| Lang | Mean length of disfluent sentences | Mean length of fluent sentences |
|------|------|------|
| En | $9.19 \pm 2.85$ | $7.45 \pm 2.59$ |
| Hi | $10.18 \pm 3.60$ | $8.24 \pm 3.12$ |
| De | $7.25 \pm 3.12$ | $5.71 \pm 2.84$ |
| Fr | $7.42 \pm 3.05$ | $6.08 \pm 2.87$ |

Table 4: Average length of disfluent and fluent utterances in the DISCO corpus for each language; En-English, Hi-Hindi, De-German, Fr-French.

Our corpus also contains a good distribution of sentences across various task domains. Readers are urged to refer to Appendix B for the domain-level distribution and other important plots pertaining to the corpus.

### 3.5 Helper Datasets

We also use some helper datasets, extracting unlabeled sentences to enable few shot learning-based experiments on DISCO.

**LARD:** Contains synthetically generated English disfluent sentences using rule-based disfluency injection in fluent sentences (Passali et al., 2022).

**Samanantar:** Consists of 49.7 million parallel sentences between English and 11 Indic languages (Ramesh et al., 2021). Source sentences were collected across many domains such as newspapers, government public archives, Wikipedia, etc. The corpus consists of fluent sentences, and we only use Hindi sentences for our experiments.

**GermEval 2014:** Consists of 31K German fluent sentences collected from Wikipedia and various news corpora (Benikova et al., 2014). Originally used for Named Entity Recognition, we utilize unlabeled sentences from the train split.

**DiaBLa:** Released by Bawden et al. (2021), this corpus consists of 5700+ sentence pairs for English-French MT. The dataset is curated from written and informal interactions between native speakers in both languages.

## 4 Dataset Evaluation

This section describes the experiments we perform to evaluate the DISCO corpus. Our evaluation strategy measures the efficacy of the corpus for robust disfluency correction in a wide variety of cases. Moreover, we also test the ability of our trained models to correct disfluencies for improving downstream machine translation.

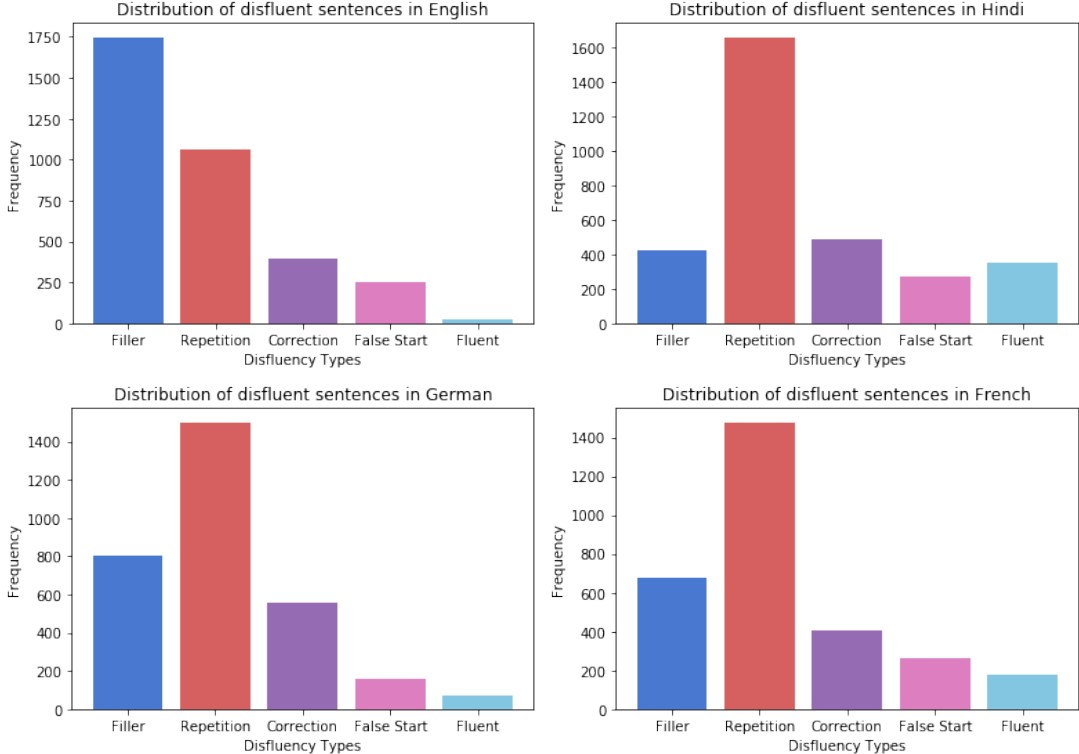

Figure 2: Distribution of sentences across disfluency types for all four languages in DISCO.

### 4.1 Data Processing

All parallel sentence pairs are passed through a punctuation removal module to reduce the number of tokens for classification. As per the structure of disfluencies described in section 3.1, we consider fluent terms to always follow disfluent terms in an utterance. Disfluent utterances are marked with the positive label (1) and fluent utterances with the neutral label (0) (Kundu et al., 2022).

### 4.2 Baseline Models

We use a combination of smaller ML models, larger transformer models and transformers with adversarial training. All models are trained on an 80:10:10 train:valid:test split for each language.

#### 4.2.1 ML Baselines

Previous work has shown the efficacy of using Conditional Random Fields (CRFs) and Recurrent Neural Network (RNN) based techniques for token classification in DC (Ostendorf and Hahn, 2013b; Hough and Schlangen, 2015b). These models require fewer labeled data and are ideal for low-resource domain-specific training (Simpson et al., 2020). Token-level features from a powerful multilingual transformer, XLM-R (Conneau et al., 2020), were used for finetuning the CRF and RNN

models.

#### 4.2.2 Transformer Baselines

Transformers (Vaswani et al., 2017) are large and powerful neural networks capable of learning complex text representations for many downstream NLP tasks. We experiment with three multilingual transformers: mBERT (Devlin et al., 2019), XLM-R (Conneau et al., 2020) and MuRIL (Khanuja et al., 2021). Finetuning for sequence tagging is performed by adding a classification head (on top of these transformers) that performs sub-word level binary prediction. Prediction of a word to be disfluent/fluent is the prediction of the first sub-word to be disfluent/fluent.

#### 4.2.3 Transformer with Adversarial Training (Seq-GAN-BERT)

In low-resource settings, adversarial training helps transformers improve the representations it learns for downstream tasks. We use the Seq-GAN-BERT model (Bhat et al., 2023), which supports adversarial training for transformers utilizing labeled and unlabeled data for token classification-based DC. Unlabeled data is used from helper datasets specified in section 3.5. We obtain the best results using MuRIL transformer as the base model in Seq-GAN-BERT.

## 4.3 Experimental Setup

CRF and RNN models are trained using the Flair-NLP framework (Akbik et al., 2019) till the validation cross-entropy loss saturates. We start with a learning rate of 0.1 and reduce it by half each time the model does not improve for three consecutive epochs. Transformer models are trained using the popular transformers package (Wolf et al., 2020). We use a learning rate of 2e-5 and a weight decay of 0.01. All transformer models are trained for 40 epochs using the Adam optimizer (Kingma and Ba, 2014).

### 4.3.1 Hardware support

All ML baselines were trained with A100 GPUs provided by Google Colab. Transformers were trained with one NVIDIA GeForce RTX P8-11GB GPU per experiment.

## 5 Results and Analysis

We thoroughly analyse all experiments performed in DC. This section also discusses some case studies highlighting strengths and weaknesses of our best models. Our experiments in analyzing the impact of DC on MT provides interesting linguistic insights into the phenomenon of disfluencies.

### 5.1 Disfluency Correction

All results are reported using the F1 score metric (Jamshid Lou and Johnson, 2017; Passali et al., 2022). Combined results across all four languages are described in table 5. As the complexity of models increases, the overall accuracy also increases. Transformer architectures perform better than CRF and RNN-based models consistently. In each language, the best models produce 90+ F1 scores on blind test sets, indicating that our corpus successfully solves the data scarcity problem. As expected, F1 scores of multilingual transformers are close due to similiar range of parameters that are fine-tuned for token classification based DC.

Performance across disfluency types is described in table 6. We observe that the model performs poorly for fluent sentences in English and French due to fewer samples in the test set. In Hindi and German, false starts are the most difficult disfluencies to correct. Further examination reveals that our model often under-corrects longer false starts, especially in the presence of other disfluencies like fillers. Our model performs robustly across all domain types of utterances. Readers are

| Model | En | Hi | De | Fr |
|---|---|---|---|---|
| CRF | 59.15 | 35.70 | 53.01 | 43.60 |
| RNN | 83.28 | 69.96 | 81.11 | 82.50 |
| mBERT | 96.94 | 88.08 | 93.70 | **92.97** |
| XLMR | 95.95 | 91.31 | **95.89** | 92.35 |
| MuRIL | 96.65 | **94.29** | 92.00 | 92.48 |
| Seq-GAN-BERT | **97.55** | 93.71 | 88.95 | 86.23 |

Table 5: Results in DC for each language. For Seq-GAN-BERT, we report best results with helper datasets (section 3.5): English (En): LARD, Hindi (Hi): Samanantar, German (De): GermEval 2014, French (Fr): DiaBLa. Since we are the first to create DC corpus in German and French and with existing English and Hindi datasets being vastly different in its properties and sources, we do not provide zero-shot metrics of our best models on other datasets.

strongly urged to refer to Appendix C for domain-level analysis of DC results. Although existing DC datasets are of diverse domains, our experiments show that models trained on DISCO outperform test sets from other DC datasets (Appendix D).

| Type | En | Hi | De | Fr |
|---|---|---|---|---|
| Filler | 99.78 | 100.00 | 99.37 | 98.72 |
| Repetition | 98.48 | 92.81 | 97.13 | 98.58 |
| Correction | 91.72 | 91.54 | 98.48 | 93.94 |
| False Start | 97.19 | 85.04 | 90.91 | 98.00 |
| Fluent* | 66.67 | 91.03 | 96.30 | 57.14 |

Table 6: F1 scores for every disfluency type in each language using our best DC model. *We report F1 score of fluent class here because for disfluent class, true positives is equal to zero.

Table 7 discusses some important case studies containing inferences produced by our model on unseen test sentences. Our models accurately correct complex cases such as multiple disfluencies and change in thought/speech plan. However, it also over-corrects clarifications and treats it as a correction to remove significant meaning. We observe that multi-words, a critical linguistic phenomenon in Indian languages, are often over-corrected to simplify the text. More case studies appear in Appendix E along with gloss and English transliteration for non-roman text.

| Lang | Type | Disfluent Sentence | Prediction | Comments |
|---|---|---|---|---|
| En | C | is *etrading i mean* ameritrade the top trading app currently | is ameritrade the top trading app currently | Correctly identifies change in main content |
| | R | add a 20minute 2mile *walk* walk to myfitnesspal | add a 2mile walk to myfitnesspal | Correctly removes repeated word *walk* but mistakes *2mile* as a correction to *20minute* |
| De | R,C | trage *die übung* die laufübung ein | trage die laufübung ein | Model correctly detects correction of *übung* (exercise) to *laufübung* (running exercise) |
| | FS | *zeig mir wie* sag mir wie es den aktienpreisen bei der nasdaq gestern ging | *zeig mir wie* sag mir wie es den aktienpreisen bei der nasdaq gestern ging | Model fails to remove false started phrase *zeig mir wie* as it has an independent meaning |
| Fr | F,R | envoyez *mon message euuh* mon message audio | envoyez mon message audio | Model correctly identifies that the user repeats the phrase to denote an audio message. |
| | F | enregistrer une vidéo hd avec *hmm* instagram | enregistrer une vidéo avec instagram | Model incorrectly thinks *hd* is a disfluent term and not the abbreviation of High Definition |
| Hi | C | कैब *बुक नहीं* कैंसिल कर दो कल शाम की दिल्ली के लिए ओला पर | कैब कैंसिल कर दो कल शाम की दिल्ली के लिए ओला पर | Chain in thought process detected & corrected |
| | R | फ़ोन पर *आधा घंटा* आधा घंटा फ़ास्ट रनिंग शुरू कर दो अभी के अभी | फ़ोन पर आधा घंटा फ़ास्ट रनिंग शुरू कर दो अभी | Model removes repetition but also over corrects the multi-word phrase: अभी के अभी |

Table 7: Inference examples from DC models; En-English, De-German, Fr-French, Hi-Hindi; F-Filler, R-Repetition, C-Correction and FS-False Start.

| Lang Pair | Setup | F | R | C | FS |
|---|---|---|---|---|---|
| Hi-En | No DC | 31.79 | 41.45 | 29.37 | 38.67 |
| | ADC | 42.34 | 47.92 | 35.94 | 45.54 |
| | HDC | 42.96 | 47.69 | 38.71 | 45.23 |
| De-En | No DC | 37.95 | 39.36 | 38.95 | 57.31 |
| | ADC | 51.20 | 48.50 | 50.76 | 67.35 |
| | HDC | 51.20 | 48.63 | 51.32 | 68.40 |

Table 8: Effect of each disfluency type and its removal on downstream MT for Hindi-English (Hi-En) and German-English (De-En) language pairs. F-Filler, R-Repetition, C-Correction and FS-False Start.

## 5.2 Impact of Disfluency Correction on Downstream Machine Translation

We use a strong baseline NLLB MT system (Costa-jussà et al., 2022) to compare English translations produced with and without disfluency removal (Appendix F) to understand the impact of DC mod-els on an important downstream NLP task.

The Ground Truth (GT) translations for Hindi-English and German-English were created by respective language experts. We use the sacrebleu package (Post, 2018) to calculate BLEU scores between: T1 (Translations without DC) and GT; T2 (Translations with Automatic DC) and GT; and T3 (Translations with Human DC) and GT. Table 10 summarises our results in both language pairs. DC improves downstream MT for Hindi-English by **6.44** points and for German-English by **4.85** points in BLEU score. We also observe that human DC outperforms automatic DC, highlighting scope of improvement of DC models.

Table 8 shows that translation BLEU score improves for every disfluency type after DC. Moreover, in repetition and false starts, the automatic removal of DC slightly outperforms Human DC. The most significant improvement in BLEU score is observed in fillers, with the lowest improvement in corrections. Our models also improve the scores

| Lang Pair | Disfluent Sentence | Predicted Fluent Sentence (ADC) | Translations | Observations |
|---|---|---|---|---|
| De-En | 30 nein 50 minuten joggen starten | 50 minuten joggen starten | **T1:** 30 no 50 minutes start jogging
**T2:** Start running for 50 minutes
**T3:** Start jogging for 50 minutes. | Correction from 30 to 50 minutes is identified by DC which leads to fluent translations T2 and T3 |
| | mache eine weitwinkel also eine weitwinkel video | mache ein video | **T1:** So make a wide angle video
**T2:** Make a video
**T3:** Make a wide angle video | ADC mistakenly removes both utterances of *weitwinkel* (wide angle) leading to downstream translation error |
| Hi-En | वाच में रनिंग अह्ह्ह स्मार्ट वॉच में रोका हुआ रनिंग टाइमर रिज्यूम करो | स्मार्ट वॉच में रोका हुआ रनिंग टाइमर रिज्यूम करो | **T1:** Running in the Watch Ahh Smart Watch Restart the Running Time that has been blocked in the Watch
**T2:** Resume the running timer that is blocked in the smartwatch
**T3:** Resume stopped running timer in smart watch. | T1 is difficult to interpret due to the presence of translated false start phrase. |
| | 5 मील का दौड़ना एक अ अ अ बस अब बंद कर दो | 5 मील का दौड़ना एक बस अब बंद कर दो | **T1:** Running 5 miles a a a a just stop now
**T2:** 5 miles running a bus now stop
**T3:** Stop running 5 miles now. | Due to incomplete disfluency correction, dubious translation (bus or stop) of the following word causes an issue: बस |

Table 9: Examining some examples where disfluencies impact machine translation output for German-English (De-En) and Hindi-English (Hi-En) language pairs

| Setup | Hi-En | De-En |
|---|---|---|
| MT without DC | 36.19 | 37.08 |
| MT with ADC | 42.63 | 41.93 |
| MT with HDC | 43.52 | 42.10 |

Table 10: Effect of DC on downstream MT for Hindi-English (Hi-En) and German-English (De-En) language pairs. ADC: Automatic Disfluency Correction, HDC: Human Disfluency Correction

| Model (Dataset) | BLEU Score |
|---|---|
| MuRIL (DISCO) | 42.63 |
| MuRIL (Kundu et al., 2022) | 38.97 |

Table 11: Comparing the performance of DC systems trained on different datasets on the Hindi - English DISCO MT improvement task when used with a state-of-the-art MT system (NLLB)

across all domains, as described in Appendix F.

We also compared the downstream MT improvement caused by a large transformer (MuRIL) trained separately on both DISCO and (Kundu et al., 2022) for Hindi DC followed by downstream Hindi - English translation using NLLB. Table 11 highlights that MuRIL trained on DISCO leads to a **3.66** BLEU score improvement relative to the baseline.

### 5.2.1 Case Study: Examining a few translations with and without disfluency correction

Table 9 discusses some interesting cases where disfluencies impact the English translation of Hindi and German sentences. Although removing disfluencies in most cases helps MT, there are few examples where DC leads to worse output.

## 6 Conclusion and Future Work

This paper introduces the DISCO dataset for disfluency correction in four widely spoken languages:

English, Hindi, German and French. Our work highlights the importance of large-scale projects in NLP that scale the amount of labeled data available. Spoken interactions between humans and AI agents are riddled with disfluencies. Eliminating disfluencies not only improves readability of utterances but also leads to better downstream translations. Our dataset, which consists of roughly 3000 parallel disfluent-fluent sentences in each language, significantly reduces the data scarcity problem in DC. This allows training of large transformer models to correct spoken disfluencies from written transcripts with high accuracy. Lack of conversational translation datasets has led to most MT systems trained on fluent text. Our experiments show that such models if used in conversational settings do not perform well. By adding a DC model in the pipeline, which is often a smaller model with an incremental increase in latency, one can improve the downstream translations outputted by an MT system that does not adjust to conversational phenomena. Moreover, our dataset in German - English and Hindi - English can also be used to finetune conversational MT models.

Future work lies in experimenting with better ML models for sequence tagging-based DC supporting multilingual training. These should also incorporate linguistic features like reparandum, interregnum and repair. Multimodal DC presents a promising direction as it has the capability of using both speech and text features for correction tasks (Zhang et al., 2022). Additionally, trained DC models must be evaluated using diverse samples from various domains and dialects. Special efforts must be made to collect disfluent speech transcripts to be annotated and trained for DC in other low-resource languages.

## 7 Acknowledgements

We would like to thank the anonymous reviewers and area chairs for their suggestions to strengthen the paper. This work was done as part of the Bahubhashak Pilot Project on Speech to Speech Machine Translation under the umbrella of National Language Technology Mission of Ministry of Electronics and IT, Govt. of India. We would also like to thank the project managers, internal and external language translators at the Computation for Indian Language Technology (CFILT) IIT Bombay.

## Limitations

Our work consists of two limitations. Firstly, since our annotation process consisted of one annotator for each language, we could not report metrics such as inter-annotator agreement or Cohen's kappa to prove the validity of our dataset. However, since DC is a relatively more straightforward task and consists of only removing disfluent words from spoken utterances, the authors were able to verify many samples as a part of their analysis. Moreover, the structure of disfluencies helps us recognize disfluency types easily. We have also provided a few flagged cases where annotators discussed their queries with us and how we resolved them.

Secondly, we do not compare trained models on DISCO with other datasets due to varied domain of existing datasets. We found that existing datasets like Switchboard (Godfrey et al., 1992), LARD (Passali et al., 2022) and Kundu et al. (2022) all consisted of utterances from very diverse data sources. However we include experiments in Appendix D that highlight the robustness of models trained on DISCO.

## Ethics Statement

This work publishes a large scale human annotated dataset for disfluency correction in 4 Indo-European languages - English, Hindi, German and French. We have taken all steps to ensure that the data is collected and annotated using all ethical means. The source sentences of our dataset are extracted from Goel et al. (2023) which release the data using the CC by 4.0 license, allowing us to remix, transform, and build upon the material for any purpose. We also follow a stringent data annotation protocol with consent from the annotators and ensuring they are aware of the risks associated with data creation. We also mention the compensation paid to them for their contribution in section 3.3. Since this project is not sponsored by a federal body, we do not use the IRB approval for our work. However, attention is paid to the quality of our dataset with flagged cases discussed extensively with annotators to ensure appropriate resolution (Appendix A). A thorough and extensive analysis of our corpus is performed, details of which are provided in section 3. All datasets used in conjunction with our corpus are open-source and cited appropriately in section 3.5. We understand that the dataset might have some mistakes, and we

will continuously work on monitoring and resolving such issues once the corpus is published for open-source research. Our robust results across domains and types of sentences ensure that changes to the dataset do not pose any technical issues or risks to both developers and model users.

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

## A   Flagged Cases in Data Creation

Data in DC is expensive and challenging to annotate. Language experts must not only have complete knowledge of the grammar and semantics of a language, but they should also be mindful of the disfluency structure and rules for token classification. In our work, we could only assign one annotator per language. To ensure that the data created is high quality, we had regular discussions with our annotators to resolve flagged examples. Table 12 showcases such examples with the reasoning behind our data annotation decisions. After verifying the quality of data created, we proceed with data analysis as described in section 3.4.

## B   Additional Dataset Analysis

In this section, we provide information about the origin of the utterances that are part of the DISCO dataset. We use domain type labels from Goel et al. (2023). To better understand our data, we describe each domain type under broader categorization. In each example, disfluent utterances are marked in red. The number of sentences in each domain type for each language is specified in table 13.

- **Health Fitness:** Sentence pairs belonging to this domain consist of utterances where the user wants to perform a fitness or health checkup task like recording his/her exercises, nutrition or blood sugar. Any interaction where the user discusses any fitness query can be tagged in this category. Domains such as Get health stats, Log exercise, Log nutrition, Start exercise, Stop exercise, Pause exercise and Resume exercise fall under this type.

  Example - Go to Fitbit and show me my um my blood sugar reading

- **Order Status:** Sentence pairs belonging to this domain consist of utterances where the user wants to check the status of the already placed order. The Check order status domain falls under this type.

  Example - Check the status of um of my Poshmark order with FedEx.

- **Finance :** Sentence pairs belonging to this domain consist of utterances where the user wants to perform a finance task like checking stock market prices or getting information from a finance app. The domain Get security price falls under this type.

  Example - I want to um check stock prices.

- **Bill Payment or Purchase:** Sentence pairs belonging to this domain consist of utterances

where the user wants to complete a bill payment or is instructing the AI agent to purchase something for him/her. Domains such as Get bill, Pay bill, Get product, BuyEventTickets, GetGenericBusinessType, Order menu item fall under this type.

Example - Pay my um my phone bill for this month.

- **Internal Task:** Sentence pairs belonging to this domain consist of utterances where the user wants the AI agent to perform a task which does not involve any extra application. Examples could be sending a message/email to someone, cancelling some plan, taking some notes, etc. If a third-party application is used in the utterance, it is an "External Task"; if not, it is an "Internal Task". Domains such as Get message content, Add contact, Create note, Open app, Take photo, Add item to list fall under this type.

  Example - I want to e-mail Zane this photo and cc um and cc Zach.

- **External Task:** Sentence pairs belonging to this domain consist of utterances where the user wants the AI agent to perform a task with the help of a third-party application. In this domain, you will find utterances where the user specifies the AI agent and which application should the AI agent use to complete the task. Domains such as Cancel ride, Order ride, Post message fall under this type.

  Example - Use WhatsApp to to send location to Jim.

We also depict the word cloud of disfluent sentences across the four languages. Our analysis shows the most common disfluent words across four languages. Since the Filler class occupies a majority in the distribution, for each language, we see filler words like um, uh, er, and umm occupy a considerable size in the cloud for English. Similarly, common fillers in Hindi, German and French are the biggest in the respective word clouds (figure 3).

Correlation analysis between original Hindi and German sentences and their respective English translations was also performed to ensure that the number of outliers was minimum and the slope of points followed a natural straight line. Figure 4 depicts the straight-line scatter plots observed.

The disco corpus contains a good representation of shorter and longer disfluent sentences across each language, increasing the complexity of corrections needed. Figure 5 depicts the box plot of disfluent sentences, indicating the average sentence lengths of spoken utterances across four languages. These plots summarize our analysis and motivate us to test this dataset across various ML models (section 4).

# C  Domain Level Analysis of DC Results

We show the domain-wise performance of our DC models in each language in table 14. The best models can reach 100 per cent accuracy for many domain types. However, there are still some domains, such as Log exercise, Get Bill and Log nutrition, where the performance varies significantly for each language. Our results show robust performance across domain as well as disfluency types (section 5.1).

# D  Evaluating models trained on DISCO using other DC datasets

Models trained on DISCO outperformed test sets from other DC datasets. Table 15 shows our experiments in testing a large transformer model trained on the DISCO dataset with other open-source test sets. We ensured equal distribution across disfluency types while creating the test sets from DISCO, LARD (Passali et al., 2022) and Switchboard (Godfrey et al., 1992) datasets. MuRIL was used as the transformer for our experiments because it reported a high F1 score across other language models. Moreover, since labeled datasets are being evaluated, we could not use the Seq-GAN-BERT model as it required unlabeled data. Our experiments indicate that models trained on the DISCO dataset, perform better than models trained on real or synthetic data. Such experiments could not be performed for French and German because of lack of existing open source DC datasets in these languages.

# E  More examples of DC inference

This section shows more examples of inferences from our best models across all four languages - English (Table 16), Hindi (Table 17), German (Table 18) and French (Table 19). Since Hindi sentences are written in the Devanagari script, we provide transliteration and gloss for every example

discussed. Strong results of our models for DC motivate us to test their performance for downstream MT improvement (Section 5.2)

## F    Setup for downstream DC-MT experiments and domain level results

Disfluency Correction has been studied predominantly as post editing task for downstream tasks. In this section, we discuss important experiments in understanding the impact of disfluencies for downstream machine translation. We work with the Hindi - English and German - English language pairs. We use the NLLB MT system ((Costa-jussà et al., 2022)) for our experiments. The following steps are followed -

Disfluency Correction has been studied predominantly as a post-editing task for downstream problems. This section discusses essential experiments in understanding the impact of disfluencies on downstream machine translation. We work with the Hindi - English and German - English language pairs. Our experiments use the NLLB MT system ((Costa-jussà et al., 2022)). The following steps are followed -

- Disfluent sentences are passed through the MT system to create translations T1

- Automatic Disfluency Corrected sentences (using our best DC models) are passed through the MT system to create translations T2

- Human-corrected sentences (as provided by our annotators) is passed through the MT system to create translations T3

- We denote ground truth translations as GT. T1 is compared with GT to calculate BLEU_DIS. Similarly, BLEU_ADC is the score between T2 and GT, and BLEU_HDC is the score between T3 and GT.

- These three BLEU scores indicate the performance of machine translation in the absence of disfluency correction BLEU_DIS, in the presence of automatic disfluency correction BLEU_ADC and the presence of human disfluency correction BLEU_HDC

Domain level results for Hindi-English and German-English sentences and changes in BLEU score observed with and without DC are given in table 20 and table 21, respectively. Some important examples where DC impacts downstream MT is discussed in section 5.2.1.

| Lang | Disfluent Sentence | Confusion and Resolution |
|---|---|---|
| En | I need to construct or create a new note for andy | *construct* a note does not make sense and *create* could be a correction |
| | | The utterance does not indicate any change in speech plan, this could be a semantic mistake made by a non-native speaker. Output should be the same as input due to fluent sentence |
| | I want to e-mail Zane this photo and cc um and cc Zach | This example contains both repeated phrase *and cc* and filler *um*. What should be the correct disfluency type label assigned? |
| | | We observe that the speaker hesitates at the utterance *and cc* at which point the speech plan is changed. Thus, repetition plays a more important role here |
| Hi | राम ओह नहीं सीमा को एक वौइस् मैसेज भेजो | The second word is a filler and must be removed. However, should we remove the first word too because we can lose out on information? |
| | | In this utterance, the first word is a part of reparandum and hence needs to be removed too. |
| | मैं अपने कमर ह्म्म कमर की साईज जानना चाहता हूँ | This sentence contains repetition and filler. What should we mark as disfluency type? |
| | | In this example, repetition plays a greater role in introducing the disfluency |
| De | Brauche Sandalen von äh Kick. | In this example, is the word *Kick* uttered as a correction? |
| | | No, Kick is the name of a shop in Germany |
| | Pause, drück auf Pause. | The word *pause* is repeated, but is not consecutive. What should the label be? |
| | | In this example it seems the user uttered *pause* but then corrected himself to give more information about what he wants to pause. Correction tag seems most applicable |
| Fr | mets-moi jouons à Fortnite | *Mets-moi à Fortnite* OR *jouons à Fortnite* are the gramatically correct options. |
| | | In this example, it seems the speaker uttered the false started phrase *mets-moi* first and then corrected themselves. Considering it to be a False Start, the first part must be removed. |
| | Annulation ation du Uber. | Should the tag be Correction or Filler since *ation* does not mean a sound/noun |
| | | *ation* seems like a mistaken utterance and hence should be tagged as correction |

Table 12: Flagged examples in data creation and our reasoning to create fluent sentences and disfluency labels

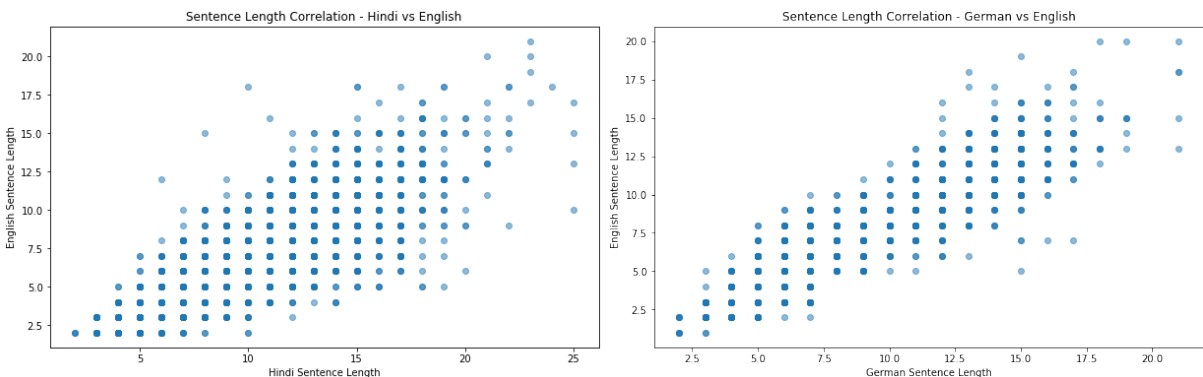

Figure 3: Word cloud of disfluent sentences across each language in the DISCO corpus, showcasing the most common disfluent words observed in spoken utterances

Figure 4: Plotting the correlation between disfluent sentences and their English translations. These graphs indicate that the number of words in any English translation of Hindi or German sentences can be estimated using a straight line slope as depicted with minimum outlier cases.

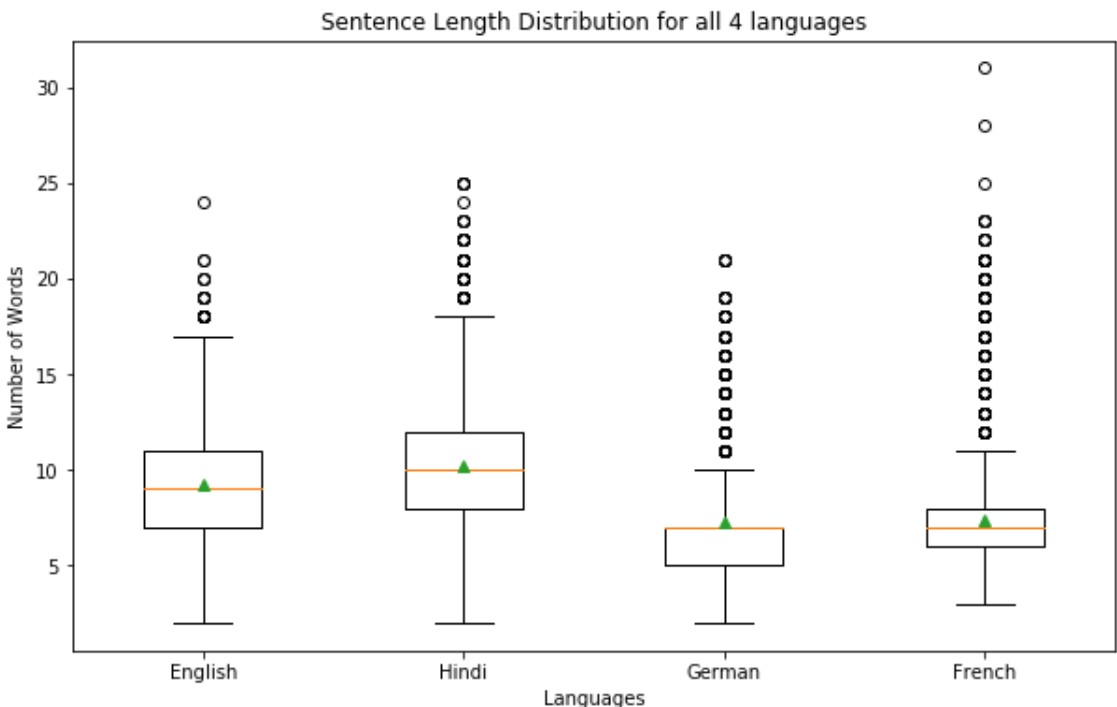

Figure 5: Box plot of disfluent sentence lengths across all languages in DISCO corpus

| Domain Type | English | Hindi | German | French |
|---|---|---|---|---|
| Send digital object | 222 | 116 | 116 | 121 |
| Get health stats | 258 | 105 | 140 | 95 |
| Get message content | 191 | 68 | 69 | 100 |
| Add contact | 286 | 143 | 106 | 118 |
| Create note | 68 | 62 | 80 | 45 |
| Check order status | 274 | 150 | 90 | 72 |
| Get bill | 255 | 131 | 82 | 68 |
| Get security price | 238 | 117 | 85 | 68 |
| Open app | 238 | 156 | 185 | 226 |
| Pay bill | 251 | 128 | 109 | 105 |
| Get product | 218 | 94 | 116 | 98 |
| Other | 223 | 72 | 89 | 62 |
| Post message | 256 | 134 | 178 | 169 |
| Record video | 25 | 129 | 152 | 154 |
| Log exercise | 228 | 129 | 128 | 73 |
| Log nutrition | 248 | 88 | 98 | 87 |
| Take photo | 0 | 108 | 140 | 150 |
| Cancel ride | 0 | 191 | 109 | 170 |
| Order ride | 0 | 138 | 86 | 148 |
| BuyEventTickets | 0 | 98 | 118 | 85 |
| Play game | 0 | 115 | 133 | 151 |
| GetGenericBusinessType | 0 | 103 | 101 | 38 |
| Start exercise | 0 | 164 | 151 | 133 |
| Stop exercise | 0 | 172 | 170 | 164 |
| Pause exercise | 0 | 129 | 157 | 100 |
| Resume exercise | 0 | 140 | 108 | 165 |
| Order menu item | 0 | 0 | 0 | 25 |
| Add item to list | 0 | 0 | 0 | 15 |

Table 13: Language wise distribution of domain of disfluent sentences in DISCO corpus

| Domain Type | English | Hindi | German | French |
|---|---|---|---|---|
| Send digital object | 97.37 | 97.56 | 100.0 | 100.0 |
| Get health stats | 98.55 | 92.68 | 100.0 | 100.0 |
| Get message content | 98.36 | 85.71 | 90.0 | 100.0 |
| Add contact | 97.56 | 86.49 | 100.0 | 100.0 |
| Create note | 89.66 | 100.00 | 82.35 | 100.0 |
| Check order status | 98.08 | 93.88 | 95.65 | 100.0 |
| Get bill | 100.00 | 57.78 | 100.0 | 92.31 |
| Get security price | 100.00 | 91.89 | 84.21 | 90.91 |
| Open app | 100.00 | 97.87 | 92.31 | 100.0 |
| Pay bill | 100.00 | 92.31 | 100.0 | 94.44 |
| Get product | 100.00 | 88.00 | 100.0 | 96.97 |
| Other | 100.00 | 100.00 | 100.0 | 100.0 |
| Post message | 97.30 | 93.88 | 100.0 | 100.0 |
| Record video | 100.00 | 100.00 | 96.3 | 95.83 |
| Log exercise | 94.12 | 92.00 | 100.0 | 92.31 |
| Log nutrition | 98.31 | 66.67 | 100.0 | 100.0 |
| Take photo | - | 100.00 | 100.0 | 96.15 |
| Cancel ride | - | 98.51 | 90.0 | 100.0 |
| Order ride | - | 94.87 | 98.04 | 100.0 |
| BuyEventTickets | - | 92.68 | 100.0 | 100.0 |
| Play game | - | 94.44 | 94.92 | 97.67 |
| GetGenericBusinessType | - | 94.12 | 100.0 | 75.0 |
| Start exercise | - | 87.50 | 97.96 | 96.55 |
| Stop exercise | - | 94.74 | 100.0 | 98.11 |
| Pause exercise | - | 85.11 | 93.33 | 85.71 |
| Resume exercise | - | 94.12 | 100.0 | 90.0 |
| Order menu item | - | - | - | 100.0 |

Table 14: F1 scores for disfluency correction for every domain type in each language

| Language | Test Dataset | Model Name | Test F1 |
|---|---|---|---|
| English | DISCO | MuRIL finetuned on DISCO | 96.65 |
| | | MuRIL finetuned on Switchboard | 86.97 |
| | | MuRIL finetuned on LARD | 89.33 |
| | Switchboard | MuRIL finetuned on DISCO | 95.67 |
| | | MuRIL finetuned on Switchboard | 88.44 |
| | | MuRIL finetuned on LARD | 89.33 |
| | LARD | MuRIL finetuned on DISCO | 96.21 |
| | | MuRIL finetuned on Switchboard | 91.23 |
| | | MuRIL finetuned on LARD | 97.30 |
| Hindi | DISCO | MuRIL finetuned on DISCO | 94.29 |
| | | MuRIL finetuned on Kundu et al. (2022) | 81.16 |
| | (Kundu et al., 2022) | MuRIL finetuned on DISCO | 87.16 |
| | | MuRIL finetuned on Kundu et al. (2022) | 84.81 |

Table 15: Test F1 scores of a transformer based model trained on the DISCO dataset compared to other open-source datasets. For LARD, our model performs comparatively even in out-of-domain synthetic test sets.

| Lang | Type | Disfluent Sentence | Predicted Fluent Sentence | Comments |
|---|---|---|---|---|
| En | F,R | ask chase how much do uh how much my visa bill is | ask chase how much my visa bill is | Able to remove more than one type of disfluency numerical disfluency identification |
| | C | put 203 wait 2086683553 into my contacts | put 2086683553 into my contacts | |
| | C | text sal at 4556329902 actually 4556329092 | text sal at 4556329092 | |
| | FS | ब्लैक फ्राइडे मेरा मतलब है साइबर मंडे पर सेल फ़ोन के सर्वश्रेष्ठ डील क्या है | साइबर मंडे पर सेल फ़ोन के सर्वश्रेष्ठ डील क्या है | Utterance had complete change in content, detected correctly |
| | C | lets record on fitbit that i did 10 pushups wait make that 20 pushups this morning | lets record on fitbit that i did make that 20 pushups this morning | Incomplete correction |
| | FL | i need to construct or create a new note for andy | i need to create a new note for andy | Tradeoff between correction and content preservation |
| | F,R | make my status on facebook um make it anyone down for a movie | make my status on facebook make it anyone down for a movie | Issues when more than one type of disfluency is present |
| | F,C | add the number 5551987 to my husbands to his umm work contacts list | add the number 5551987 to my husbands to his work contacts list | example where normal syntactic structure of disfluencies is not followed |
| | C | what was my water in in intake last month | what was my water in intake last month | Model mistakes correction as repetition |

Table 16: Inference examples from English DC models; En - English

| Lang | Type | Disfluent Sentence | Transliteration | Gloss | Predicted Fluent Sentence | Comments |
|---|---|---|---|---|---|---|
| Hi | C | पिछले महीने की मेरी उम्म कार की कितनी ई अअ ईएमआई बकाया है | Pichhale mahīne kī merī um-mma kār kī kitanī ī aa īemaāī bakāyā hai | last month did my ummm car did howmany e aa emi arrears is | पिछले महीने की मेरी कार की कितनी ईएमआई बकाया है | Able to remove multiple occurrences of fillers |
| | R | एकिनेटर का लेवल 4 शुरू कर दो कर दो शुरू | Ekineṭar kā leval 4 shurū kar do kar do shurū | akinator Of level 4 start do two do Two start | एकिनेटर का लेवल 4 शुरू कर दो | Solves ambiguity to maintain grammar |
| | C | कैब बुक नहीं कैंसिल कर दो कल शाम की दिल्ली के लिए ओला पर | Kaib buk nahīan ka-iansil kar do kal shām kī dillī ke lie olā para | cab book No cancel do it tomorrow evening did delhi of for ola on | कैब कैंसिल कर दो कल शाम की दिल्ली के लिए ओला पर | Chain in thought process detected & corrected |
| | FS | ब्लैक फ्राइडे मेरा मतलब है साइबर मंडे पर सेल फ़ोन के सर्वश्रेष्ठ डील क्या है | Blaik frāiḍe merā mata-lab hai sāibar manḍe par sel ☐on ke sar-vashreṣṭha ḍīl kyā hai | black friday My Meaning Is cyber Monday on Cell phone of the best deal what Is | साइबर मंडे पर सेल फ़ोन के सर्वश्रेष्ठ डील क्या है | Utterance had complete change in content, detected correctly |
| | F,C | डॉमिनोज से एक अम्म दो मीडियम एक्स्ट्रा चीज स्वीट कॉर्न पिज़्ज़ा मेरे अअअ डेबिट कार्ड से लो | Ḍa☐minoj se ek am-mma do mīḍiyam eksṭrā chīj svīṭ ka☐rna pijajaā mere aaa ḍebiṭ kārḍa se lo | Domino's From One mmmm Two medium extra Thing sweet corn pizza My AAA debit card From Take | डॉमिनोज से एक दो मीडियम एक्स्ट्रा चीज स्वीट कॉर्न पिज़्ज़ा मेरे डेबिट कार्ड से लो | Model removes filler but does not make correction |
| | R | फ़ोन पर आधा घंटा आधा घंटा फ़ास्ट रनिंग शुरू कर दो अभी के अभी | fon par ādhā ghanṭā ādhā ghanṭā ☐āsṭa rani-anga shurū kar do abhī ke abhī | Phone On half Bell half Bell fast running Start do Two Now Of Now | फ़ोन पर आधा घंटा फ़ास्ट रनिंग शुरू कर दो अभी | Model removes repetition but also treats the following phrase as a possible disfluent utterance: अभी के अभी |

Table 17: Inference examples from Hindi DC models; hi - Hindi

| Lang | Type | Disfluent Sentence | Predicted Fluent Sentence | Comments |
|---|---|---|---|---|
| De | F | ich muss um *öhh* 1830 am flughafen sein | ich muss um 1830 am flughafen sein | Filler *öhh* correctly identified and corrected. *um* is not a disfluent word in German and the model correctly recognizes this. |
| | F,R | die telefonrech-nung *vom emmm* vom letzten monat über paypal bezahlen | die telefonrech-nung vom letzten monat über paypal bezahlen | *vom* is repeated with the filler *emmm* in between which causes ambiguity regarding the phone bill (*telefonrechnung*). Model correctly identifies and removes the disfluent utter-ances |
| | FS | *mache ähm* kannst du ein video in hd machen | kannst du ein video in hd machen | Speaker starts with the phrase *mache ähm* but changes speech plan to ut-ter the latter fluent phrase. Model correctly removes false started phrase |
| | R | hey fedex zeige mir den status von den *status der* bestellung mit der nummer 111222abc | hey fedex zeige mir den status von sta-tus der bestellung mit der nummer 111222abc | *den status* is repeated in the original phrase but the model only removes one of the repeated words, out-put remains disfluent. In this case, repetitions are not consecutive and thus relatively difficult for the model to correct |
| | R,C | übung *für* auf strava *übung auf* pause | übung für auf strava | example where normal syntactic structure of dis-fluencies is not followed |

Table 18: Inference examples from German DC models; De - German

| Lang | Type | Disfluent Sentence | Predicted Fluent Sentence | Comments |
|------|------|--------------------|----------------------------|----------|
| Fr | F | Ouvrir euh l'application Facebook sur mon Iphone. | ouvrir lapplication facebook sur mon iphone | Filler *euh* correctly detected and removed by model |
| | F,R | Envoyez mon message euuh mon message audio. | envoyez mon message audio | User repeats the phrase *mon message* but the second time they mention that it is an audio message. Hence the earlier repeated phrase is removed along with the filler word *euuh* |
| | C | Trouver des remises des infos sur les remises. | trouver des infos sur les remises | User asks for discounts and then corrects by asking for discounts information. Model correctly detects this |
| | FS | Annule, euh, tu peux annuler la course. | tu peux annuler la course | User starts phrase to cancel, then corrects to ask to cancel the course. Model identifies the false started phrase and removes it |
| | C | J'aimerais relancer relance mon entraînement HIIT. | jaimerais relancer relance mon entraînement hiit | *relancer* is corrected to *relance* but the model doesnt detect this |
| | F | Enregistrer une vidéo HD avec hmm Instagram. | enregistrer une vidéo avec instagram | Model incorrectly thinks *hd* is a disfluent term |

Table 19: Inference examples from French DC models; fr - French

| Domain Type | MT without DC | MT with ADC | MT with HDC |
|---|---|---|---|
| Send digital object | 49.17 | 56.20 | 53.67 |
| Get health stats | 28.05 | 30.77 | 34.69 |
| Get message content | 27.62 | 28.72 | 26.42 |
| Add contact | 32.29 | 48.34 | 41.81 |
| Create note | 42.18 | 47.63 | 47.63 |
| Check order status | 44.03 | 45.50 | 46.96 |
| Get bill | 44.80 | 57.29 | 47.33 |
| Get security price | 39.91 | 51.98 | 52.50 |
| Open app | 43.40 | 50.08 | 51.65 |
| Pay bill | 41.38 | 49.83 | 50.16 |
| Get product | 31.37 | 35.25 | 33.37 |
| Other | 22.39 | 22.58 | 22.58 |
| Post message | 42.46 | 48.56 | 48.66 |
| Record video | 50.34 | 62.23 | 62.23 |
| Log exercise | 30.92 | 29.37 | 29.02 |
| Log nutrition | 26.21 | 31.34 | 33.08 |
| Take photo | 44.13 | 42.38 | 42.38 |
| Cancel ride | 36.29 | 42.36 | 45.14 |
| Order ride | 33.77 | 41.93 | 40.71 |
| BuyEventTickets | 27.96 | 23.43 | 24.67 |
| Play game | 30.04 | 42.11 | 47.67 |
| GetGenericBusinessType | 32.92 | 37.16 | 42.32 |
| Start exercise | 26.91 | 48.18 | 49.56 |
| Stop exercise | 31.89 | 38.59 | 43.05 |
| Pause exercise | 32.91 | 39.95 | 47.76 |
| Resume exercise | 33.67 | 36.96 | 36.96 |

Table 20: Effect of disfluency correction on downstream machine translation across different domain of sentences for Hindi - English pair

| Domain Type | MT without DC | MT with ADC | MT with HDC |
| --- | --- | --- | --- |
| Send digital object | 56.01 | 87.20 | 87.20 |
| Get health stats | 41.56 | 53.79 | 53.79 |
| Get message content | 15.59 | 34.62 | 34.62 |
| Add contact | 33.94 | 41.86 | 41.86 |
| Create note | 35.83 | 52.92 | 52.56 |
| Check order status | 49.79 | 59.07 | 61.56 |
| Get bill | 49.52 | 58.31 | 58.31 |
| Get security price | 34.71 | 37.99 | 40.11 |
| Open app | 49.71 | 61.38 | 61.38 |
| Pay bill | 44.70 | 73.85 | 73.85 |
| Get product | 31.64 | 33.25 | 33.25 |
| Other | 36.75 | 44.12 | 44.12 |
| Post message | 46.63 | 72.00 | 72.00 |
| Record video | 38.29 | 46.65 | 46.65 |
| Log exercise | 30.76 | 40.70 | 40.70 |
| Log nutrition | 41.90 | 37.42 | 37.42 |
| Take photo | 41.63 | 38.45 | 38.45 |
| Cancel ride | 48.23 | 54.91 | 54.91 |
| Order ride | 24.55 | 42.42 | 42.42 |
| BuyEventTickets | 44.55 | 44.31 | 44.31 |
| Play game | 36.77 | 59.36 | 59.36 |
| GetGenericBusinessType | 43.64 | 51.06 | 51.06 |
| Start exercise | 45.30 | 48.37 | 48.37 |
| Stop exercise | 26.94 | 21.49 | 21.49 |
| Pause exercise | 31.57 | 36.14 | 38.33 |
| Resume exercise | 51.39 | 46.89 | 46.89 |

Table 21: Effect of disfluency correction on downstream machine translation across different domain of sentences for German - English pair