# OpenReview forum: "DISCO: A Large Scale Human Annotated Corpus for Disfluency Correction in Indo-European Languages"
_EMNLP/2023/Conference — EMNLP 2023 Findings_

### Official Review · Reviewer_3wc9 · 2023-07-28

**Soundness:** 3

**Excitement:**

3: Ambivalent: It has merits (e.g., it reports state-of-the-art results, the idea is nice), but there are key weaknesses (e.g., it describes incremental work), and it can significantly benefit from another round of revision. However, I won't object to accepting it if my co-reviewers champion it.

**Paper Topic And Main Contributions:**

- This paper introduces and describes the creation of a disfluency correction dataset, named *DISCO*. The dataset comprises ~12k samples distributed across four languages: English, Hindi, German, and French, with about ~3k samples per language.
- The dataset contributes with additional disfluency data for English and Hindi but is the first for German and French.
- Utterances were sampled from an existing dataset of utterances made to AI agents (e.g. making notes, adding contacts) and annotated by a single annotator (one per language).
- Utterances are annotated in a sequence-labeling manner, with each word being associated with a binary label (fluent/disfluent) and labeled with a disfluency type (filler/repetition/correction/false start/fluent)
- Disfluency types are highly skewed with "repetition" being the majority class across all languages.
- An array of baseline systems are evaluated on the dataset and F1 scores are reported in the high 90s for classifying disfluent utterances correctly, however, results are significantly worse for fluent utterances in English and French... This is attributed to unbalanced utterance types but it is unclear how this finding should be interpreted.
- To show the downstream impact of applying disfluency correction, an MT experiment is conducted, reporting BLEU scores of an existing MT (not trained on disfluent source text) system when/when not applying disfluency correction on the Hi-En and De-En test sets. Results show that disfluency correction improves BLEU scores
- The respective test sets are translated by language experts but no further information is documented about this translation data.

**Questions For The Authors:**

- L166: How do you select sentences? Random, by hand, or using some heuristic?
- Table 3: There appears to be a very high proportion of disfluent words in the dataset. Combined with Figure 2 it appears that the dataset contains very few negative samples (fluent/non disfluent utterances). Doesn't that mean that the dataset heavily biases systems to predict words as disfluent? Could this possibly explain the outliers in Table 6?
- L196: You first write that there is only one annotator per language and then say that the *authors* take a majority vote to choose which samples to remove. Do all authors speak the four languages? How does that work?
- Figure 2: The figure displays skewed distributions across disfluency types, with certain types being barely present. How does this impact the usefulness of the dataset? Wouldn't it make sense to have a more balanced dataset?
- Section 4.2.3: What's the reason for including the Seq-GAN-BERT model? The section mentions that it might be useful in low-resource settings, however, it appears to be worse than most other systems and best when applied to English.
- Section 5.2: It's unclear what this experiment tells us. It appears straightforward that if we evaluate an MT system that is trained on clean source text, the results should be worse if evaluated on disfluent source input. Wouldn't there be more informative downstream tasks to evaluate on? E.g. intent classification.
-

**Reasons To Accept:**

The paper contributes a valuable multilingual resource for four languages. It strengthens the data foundation for English and Hindi by increasing the amount of available data, as well as acts as an enabler for German and French by now allowing disfluency correction systems to be developed and analyzed.

**Reasons To Reject:**

The paper contributes with a valuable multilingual dataset which is described in great detail. However, it is unclear how this dataset compares to existing datasets for English and Hindi - both in size and breadth (label types and coverage) as well as how it perhaps improves over challenges faced by prior datasets. Similarly, although the new languages German and French have no prior datasets to compare with, it is unclear how useful this additional data is. In particular, the results in Table 6 suggest that baselines can easily correct most disfluencies but that fluent utterances in English and French are hard to predict. A lack of analysis and comparison leaves it unclear if the dataset is trivial and how it makes a valuable addition to prior work.

**Reproducibility:**

3: Could reproduce the results with some difficulty. The settings of parameters are underspecified or subjectively determined; the training/evaluation data are not widely available.

**Reviewer Confidence:**

3: Pretty sure, but there's a chance I missed something. Although I have a good feel for this area in general, I did not carefully check the paper's details, e.g., the math, experimental design, or novelty.

---

> ### Author Rebuttal · Authors · 2023-08-28
>
> _Question 1: L166: How do you select sentences? Random, by hand, or using some heuristic?_
>
> **Author Response**
>
> For English and Hindi, we took all the sentences from the PRESTO dataset (Goel et al. (2023)). For German and French since we had budget constraints, we had to shortlist around 3000 sentences in each language. We did so by trying to maintain a constant distribution of sentences based on the length of the sentence, trying to cover shorter sentences (with easier disfluencies) to longer sentences (with more difficult disfluencies). We will include graphs depicting the sentence length distribution in the revised version of the paper.
>
> _Question 2: Table 3: There appears to be a very high proportion of disfluent words in the dataset. Combined with Figure 2 it appears that the dataset contains very few negative samples (fluent/non disfluent utterances). Doesn't that mean that the dataset heavily biases systems to predict words as disfluent? Could this possibly explain the outliers in Table 6?_
>
> **Author Response**
>
> Yes, and we have mentioned this conclusion in section 5.1 (line 345). Due to the lack of fluent sentences in our source corpus (PRESTO), the dataset we create has fewer fluent sentences. This, along with lesser number of fluent sentences in the test set, is a major reason for the skewed results in table 6. Fluent sentences are more readily available from sources such as newspapers and online websites like Wikipedia and thus can be included in appropriate experiments. Since our source sentences were from a very specific domain of human-AI interactions, we did not add fluent sentences from other datasets.
>
> _Question 3: L196: You first write that there is only one annotator per language and then say that the authors take a majority vote to choose which samples to remove. Do all authors speak the four languages? How does that work?_
>
> **Author Response**
>
> We apologize for any confusion caused because of our phrasing and we will clarify these doubts in the revised version of the paper. Since sequence tagging based DC is a relatively straightforward task, the authors used translation tools and subject knowledge to solve ambiguity. More information regarding how we deal with ambiguous cases can be found in Appendix A.
>
> _Question 4: Figure 2: The figure displays skewed distributions across disfluency types, with certain types being barely present. How does this impact the usefulness of the dataset? Wouldn't it make sense to have a more balanced dataset?_
>
> **Author Response**
>
> Skewed datasets are very common in disfluency datasets. For example, the most popular DC dataset- the Switchboard corpus has a skewed distribution towards more common disfluency types such as Fillers and Repetitions. This is attributed to the human tendency to utter these more than the other types. Although our dataset is also skewed towards these types, we also annotate a good amount of sentences in more complicated disfluencies such as false starts and corrections. We hypothesize that such contributions will be able to build DC systems that perform better across all types of disfluencies, and this is supported by our results in table 6.
>
> _Question 5: Section 4.2.3: What's the reason for including the Seq-GAN-BERT model? The section mentions that it might be useful in low-resource settings, however, it appears to be worse than most other systems and best when applied to English._
>
> **Author Response**
>
> The Seq-GAN-BERT model has shown to perform well in extremely low resource settings (where we only have around 300-400 labeled sentences). However when the number of sentences increases, we observed that the performance saturates and does not reach very high numbers, as compared to other transformer architectures. Moreover, Seq-GAN-BERT was only tried before for English and 2 other Indian languages (Hindi and Marathi) and this paper is the first work for testing it on European languages. Since Indian languages have significant differences from the chosen European languages, that could be a reason why Seq-GAN-BERT underperforms. However, general numbers for all models are in the range of >85 F1 score, which indicates that our dataset can cater to a wide variety of sequence tagging algorithms.
>
> _Question 6: Section 5.2: It's unclear what this experiment tells us. It appears straightforward that if we evaluate an MT system that is trained on clean source text, the results should be worse if evaluated on disfluent source input. Wouldn't there be more informative downstream tasks to evaluate on? E.g. intent classification._
>
> **Author Response**
>
> Lack of conversational translation datasets has led to most MT systems trained on fluent text. Our experiments show that such models if used in conversational settings do not perform well. By adding a DC model in the pipeline, which is often a smaller model with an incremental increase in latency, one can improve the downstream translations outputted by an MT system that does not adjust to conversational phenomena. Moreover, our dataset in German - English and Hindi - English can also be used to finetune conversational MT models.
>
> **Final Comments**
>
> The authors sincerely thank the reviewer for their time to assess and evaluate our submission.
>
> We apologize for the lack of clarification of these above points and would be happy to include these explanations in the revised version of this paper

---

### Official Review · Reviewer_HAs1 · 2023-08-05

**Soundness:** 3

**Excitement:**

3: Ambivalent: It has merits (e.g., it reports state-of-the-art results, the idea is nice), but there are key weaknesses (e.g., it describes incremental work), and it can significantly benefit from another round of revision. However, I won't object to accepting it if my co-reviewers champion it.

**Missing References:**

[https://arxiv.org/abs/1811.03189](https://arxiv.org/abs/1811.03189) also collects paired fluent English sentences for originally disfluent English and should likely be cited

This work is *not* the first to create labeled corpora for disfluency labeling / removal in German or French though this is stated several times in the paper: see [https://aclanthology.org/L14-1277/](https://aclanthology.org/L14-1277/), [https://arxiv.org/abs/1802.02926](https://arxiv.org/abs/1802.02926), [https://link.springer.com/chapter/10.1007/978-3-540-87391-4_35](https://link.springer.com/chapter/10.1007/978-3-540-87391-4_35), and others cited in each and citing these

**Paper Topic And Main Contributions:**

This paper constructs a disfluency tagging dataset for four languages (English, Hindi, German, and French) using ~3k sentences for each language selected from Goel et al. (2023), a dataset collecting human-AI agent dialogues for a variety of different domains.
One annotator per language tagged four types of disfluences (fillers, repetitions, corrections, false starts) and created fluent paired sentences for each sentence (?).
Several approaches are compared for disfluency classification (prediction of a disfluent or fluent label for each word) for all four languages, including CRFs and RNNs using features from XLM-R; three different multilingual encoders with classification heads attached; and Seq-GAN-BERT.
It is shown that removing disfluent-tagged words improves downstream MT performance with the NLLB model for German and Hindi (not shown for the other two languages pairs).

**Questions For The Authors:**

Question A:  How much of the data for each language has paired fluent sentences?

Question B:  It is stated that labels are predicted for each word. Model vocabularies are not mentioned in the paper but the pretrained models mentioned operate at the subword level.  Is prediction done at the subword level? How are situations where different labels are predicted for different subwords handled, a majority vote?

**Reasons To Accept:**

- New annotated datasets for disfluency tagging and potential removal in four languages (English, German, French, Hindi)
- Comparison of several models for disfluency tagging in four languages, and demonstration that the removal of disfluencies tagged using such models leads to noticeable improvements in downstream MT performance for Hindi and German to English using NLLB (which was not trained on disfluent data)
- Examples and qualitative analysis of trends

**Reasons To Reject:**

- The composition of the dataset is not clear (how many disfluent sentences have paired fluent sentences?)
- Limited comparison to past approaches and awareness of past work for French/German
- F1 appears to be reported for the classification task of fluent/disfluent, as opposed to for the different class labels, which is the slightly more challenging task typical of past work

**Reproducibility:**

3: Could reproduce the results with some difficulty. The settings of parameters are underspecified or subjectively determined; the training/evaluation data are not widely available.

**Reviewer Confidence:**

2: Willing to defend my evaluation, but it is fairly likely that I missed some details, didn't understand some central points, or can't be sure about the novelty of the work.

**Typos Grammar Style And Presentation Improvements:**

It is not clear to me how much of the data has paired fluent sentences; for example Table 3 suggests that the data is paired with disfluent-fluent sentences and Sec 3.3 says annotators were asked to create fluent sentences corresponding to disfluent utterances (as in, for each disfluent sentence there is a fluent rewrite) whereas Figure 2 shows that there fewer than 100 fluent sentences for two of the languages and fewer than 400 for the other two though there are 3k+ for each of the four languages.

Calling the task of tagging / classification 'disfluency correction' / DC rather than disfluency classification, labeling, or tagging is a little unclear in for example Sec 5.1 which it is a strict classification task and a slight departure from past work; consider rephrasing

---

> ### Author Rebuttal · Authors · 2023-08-28
>
> _Question A: How much of the data for each language has paired fluent sentences?_
>
> **Author Response**
>
> The data mentioned in table 3 specifies the amount of parallel data for every language. Thus our corpus contains 3479 disfluent-fluent sentence pairs in English, 3180 disfluent-fluent sentence pairs in Hindi, 3096 disfluent-fluent sentence pairs in German and 3005 disfluent-fluent sentence pairs in French. We will specify this clearly in the caption of table 3.
>
> _Question B: It is stated that labels are predicted for each word. Model vocabularies are not mentioned in the paper but the pretrained models mentioned operate at the subword level. Is prediction done at the subword level? How are situations where different labels are predicted for different subwords handled, a majority vote?_
>
> **Author Response**
>
> During the tokenization step, the tokenizer assigns an ID to every sub word. The first subword is marked to be labelled whereas the other subwords from the same word are ID'ed as -100 and hence ignored for prediction. Thus, the prediction of a word to be disfluent/fluent is the prediction of the first subword to be disfluent/fluent. We will include this clarification in the revised version of the paper appropriately.
>
> _Missing References:
> https://arxiv.org/abs/1811.03189 [Salesky et al. (2018)] also collects paired fluent English sentences for originally disfluent English and should likely be cited_
>
> _This work is not the first to create labeled corpora for disfluency labeling / removal in German or French though this is stated several times in the paper: see https://aclanthology.org/L14-1277/ [Cho et al. (2014)], https://arxiv.org/abs/1802.02926 [Christodoulides et al (2018)], https://link.springer.com/chapter/10.1007/978-3-540-87391-4_35 [Bove et al. (2008)], and others cited in each and citing these_
>
> **Author Response**
>
> We apologize for some of the missed references and thank the reviewer for pointing them out. We will be mentioning Salesky et al. (2018) in section 2 appropriately. However, there are some points we wanted to highlight regarding our assertion of being the first to create and share DC datasets in German and French. Although Cho et al. (2014) annotated the KIT lecture corpus for disfluencies in German, they were never shared the dataset open source. Christodoulides et al (2018) created DisMo, a tool to detect disfluencies but they use language clues to do it, and do not create a tagged disfluency dataset. Similarly, Bove et al. (2008) created a system to predict disfluencies using part of speech tags, without creating any dataset. To avoid confusion, we will mention that our work is the first “open-sourced” dataset for DC in the given languages. These past works are important to be noted and hence will be mentioned appropriately in section 2 of the revised version of this paper.
>
> _Typos Grammar Style And Presentation Improvements:
> It is not clear to me how much of the data has paired fluent sentences; for example Table 3 suggests that the data is paired with disfluent-fluent sentences and Sec 3.3 says annotators were asked to create fluent sentences corresponding to disfluent utterances (as in, for each disfluent sentence there is a fluent rewrite) whereas Figure 2 shows that there fewer than 100 fluent sentences for two of the languages and fewer than 400 for the other two though there are 3k+ for each of the four languages._
>
> _Calling the task of tagging / classification 'disfluency correction' / DC rather than disfluency classification, labeling, or tagging is a little unclear in for example Sec 5.1 which it is a strict classification task and a slight departure from past work; consider rephrasing_
>
> **Author Response**
>
> Figure 2 demonstrates the distribution of our dataset across the types of sentences - Filler, Repetition, Correction, False Start and Fluent Sentences. Here “fluent sentences” are source sentences that did not have any disfluencies, and hence were not annotated for disfluencies. Hence for every source sentence in our corpus (which might or might not have disfluencies), there is a corresponding fluent sentence.
>
> In previous literature, DC has been referred to when doing sequence tagging tasks and hence we follow the same norms. However, we will rephrase that our work is a token classification based DC in the revised version wherever appropriate.
>
> **Final Comments**
>
> The authors sincerely thank the reviewer for their time to assess and evaluate our submission.
>
> We apologize for the lack of clarification of these above points and would be happy to include these explanations in the revised version of this paper
>
> In view of the above answers, we request the esteemed reviewers to increase the score.

---

### Official Review · Reviewer_z3XW · 2023-08-10

**Typos Grammar Style And Presentation Improvements:** 1. Many places in the paper, for exam…
**Soundness:** 3

**Excitement:**

3: Ambivalent: It has merits (e.g., it reports state-of-the-art results, the idea is nice), but there are key weaknesses (e.g., it describes incremental work), and it can significantly benefit from another round of revision. However, I won't object to accepting it if my co-reviewers champion it.

**Missing References:**

1. The authors' claim that CRFs and RNN-based techniques require less labelled data and are ideal for low-resource domain-specific training is not supported by appropriate references.



**Paper Topic And Main Contributions:**

This paper presents a human-annotated Disfluency Correction (DC) dataset for Indo-European languages, including English, Hindi, German, and French, as well as some benchmark models. In this paper, the authors deal with four types of disfluencies: filler, repetition, correction, and false-start. At the end of the paper, some analysis of the benefits of DC is also discussed.

**Questions For The Authors:**

1. It is unclear why the four languages considered for the paper were chosen. There may be other low-resource Indo-European languages that require attention, so why these four?

2. It is unclear from the paper that how the helper datasets were used for few-shot learning-based experiments on DISCO.

3. Table 5 shows that the performance of Seq-GAN-BERT varies significantly across languages, and the authors' claim that this model performs well in low-resource settings via adversarial training is not adequately reflected. The lack of explanation on this table exacerbated the situation.

4. Despite having a small number of filler type data samples (as shown in Figure 2), the best DC model performs exceptionally well for this type of disfluency across all languages (as shown in Table 6). What is the reason for this? At the very least, attempting to address these issues makes the paper far more interesting.

**Reasons To Accept:**

This well-written paper provides a high-quality dataset on Disfluency Correction (DC) on four important languages, including French and German, which do not have a DC corpus other than the one proposed by the authors, as they claim. This dataset can be a good resource for improving language understanding and translation task.

**Reasons To Reject:**

1. Rather than presenting quantitative statistics in tables, the paper's results section is not well-organized and offers few insights into the impact of disfluency correction. More in-depth discussion of the performance improvements brought about by DC, as well as future directions, would be beneficial to the reader.

2. Because there are already a lot of works done in English for DC, the authors chose English as part of their corpus again, and the motivation for this is unclear. The reason for the languages considered for the work is not stated explicitly.

3. The annotation process only used one annotator per sample, resulting in a significant lack of annotation validity. Moreover, the paper does not specify whether the annotation was done by native speakers.

**Reproducibility:**

3: Could reproduce the results with some difficulty. The settings of parameters are underspecified or subjectively determined; the training/evaluation data are not widely available.

**Reviewer Confidence:**

4: Quite sure. I tried to check the important points carefully. It's unlikely, though conceivable, that I missed something that should affect my ratings.

---

> ### Author Rebuttal · Authors · 2023-08-28
>
> _Question 1: It is unclear why the four languages considered for the paper were chosen. There may be other low-resource Indo-European languages that require attention, so why these four?_
>
> **Author Response**
>
> We derive our source sentences from the PRESTO dataset (Goel et al. (2023)) which contains the following languages - English, Hindi, German, French, Spanish and Japanese. However, since we only had qualified annotators in English, Hindi, German and French, we chose these four languages. By doing so, we covered two widely spoken languages (German and French) which did not have any open source DC datasets, having different linguistic properties to provide a resource to study various types of disfluency phenomenon and its properties.
>
> _Question 2: It is unclear from the paper that how the helper datasets were used for few-shot learning-based experiments on DISCO._
>
> **Author Response**
>
> We used unlabeled disfluent/fluent sentences from the mentioned helper datasets for few shot learning (lines 298-299). The Seq-GAN-BERT model (Bhat et al. (2023)) has been used before to train DC models in extremely low resource settings using few shot learning to achieve state-of-the-art performance and thus we wanted to evaluate its performance on our DISCO dataset.
>
> _Question 3: Table 5 shows that the performance of Seq-GAN-BERT varies significantly across languages, and the authors' claim that this model performs well in low-resource settings via adversarial training is not adequately reflected. The lack of explanation on this table exacerbated the situation._
>
> **Author Response**
>
> The Seq-GAN-BERT model has shown to perform well in extremely low resource settings (where we only have around 300-400 labeled sentences). However when the number of sentences increases, we observed that the performance saturates and does not reach very high numbers, as compared to other transformer architectures. Moreover, Seq-GAN-BERT was only tried before for English and 2 other Indian languages (Hindi and Marathi) and this paper is the first work for testing it on European languages. Since Indian languages have significant differences from the chosen European languages, that could be a reason why Seq-GAN-BERT underperforms. However, general numbers for all models are in the range of >85 F1 score, which indicates that our dataset can cater to a wide variety of sequence tagging algorithms.
>
> _Question 4: Despite having a small number of filler type data samples (as shown in Figure 2), the best DC model performs exceptionally well for this type of disfluency across all languages (as shown in Table 6). What is the reason for this? At the very least, attempting to address these issues makes the paper far more interesting._
>
> **Author Response**
>
> Previous work has shown that DC models generally perform well for filler disfluencies (Kundu et al. (2022), Passali et al. (2022)) since they only contain simpler syllables like “um”, “err”, “agh”, which are easily identified by transformers as erroneous utterances. Moreover there are many sentences in all languages that have more than one type of disfluency present, with fillers being the most common additional disfluency (Appendix B: figure 3).
>
> _Missing References: The authors' claim that CRFs and RNN-based techniques require less labelled data and are ideal for low-resource domain-specific training is not supported by appropriate references._
>
> **Author Response**
>
> The performance of CRF and RNN based systems in low resource settings is studied in Simpson et al. (2020) where they use LSTM and CRF based sequence taggers for named entity recognition in low resource setups. We will include this reference in the revised version of our paper.
>
> _Typos Grammar Style And Presentation Improvements:_
>
> _Many places in the paper, for example, Tables 2 and 7, contain non-English languages without translation, which may be difficult for non-native speakers. Appropriate translations might be provided in these cases._
>
> _There are a lot of figures and tables in the appendix section, but no reference text. Their presentation is not only unappealing, but as there is no explanation for these data, their utility to readers may be deemed unnecessary._
>
> **Author Response**
>
> We apologize for the lack of English translations in table 2 and table 7, which were excluded due to space constraints. Alternatively we tried to mention appropriate examples in all languages so that the data can be understood by a wide range of readers. Due to space constraints we could not include English translations in the paper, but we will be happy to add English translations of all the examples in the revised version of our paper.
>
> We apologize for the missing explanations and captions in the appendix. We provided a forward and back reference to the main sections and appendix, but we will improve the readability of the appendix by adding more explanations in key sections and captions wherever necessary in the revised version of this paper.
>
> **Final Comments**
>
> The authors sincerely thank the reviewer for their time to assess and evaluate our submission.
>
> We apologize for the lack of clarification of these above points and would be happy to include these explanations in the revised version of this paper

---

### Official Review · Reviewer_J4JB · 2023-08-10

**Soundness:** 3

**Excitement:**

4: Strong: This paper deepens the understanding of some phenomenon or lowers the barriers to an existing research direction.

**Paper Topic And Main Contributions:**

This paper's contribution is a new disfluency correction dataset.  The dataset includes human-annotated examples from 4 languages (de,en,fr,hi) as well as translations to en for two of them (de-en,hi-en).  The authors claim that the dataset is useful with experiments showing improved disfluency correction capabilities as well as its impact on downstream NLP task performance.

**Questions For The Authors:**

Question A:  Are you able to perform some experiments like what I suggested in reasons to reject?  I think some things like--showing how your dataset improves model performance on other DC test sets, how other datasets don't perform as well on your (or their) DC test sets, and how other DC models from existing datasets don't help downstream MT as much as models trained with your dataset do--would greatly strengthen the claim that your dataset is useful.

**Reasons To Accept:**

The dataset seems useful and well-designed with respect to the collection protocol.

The authors are able to clearly show an increase in correction and downstream MT performance using their presented data.

There are some flaws discussed below, but considering the inclusion of under-represented languages in DC, it's pretty clear that the contribution is a good one.

**Reasons To Reject:**

The main thing that determines the strength of a dataset paper is how well its usefulness is demonstrated.  The authors do this by performing disfluency correction evaluation as well as downstream MT evaluation.  They show that they are able to achieve high scores, varying by language/type of disfluency.  They also show that using this dataset improves performance of MT.

The main issue here is that the authors have demonstrated well how 'disfluency correction' is useful as opposed to how *their* disfluency correction data is useful.  We see that better models perform better on the dataset for example.  This helps to validate the dataset as sound, but there are no experiments to compare to how using data that is already available performs.

Similarly for MT, they show that both human and automatic (using their data) disfluency correction improve MT, but not how *their* disfluency correction contributions improve MT.  They mention this as a limitation and future work, but I think it is a fairly basic necessity for a strong dataset paper.

**Reproducibility:**

4: Could mostly reproduce the results, but there may be some variation because of sample variance or minor variations in their interpretation of the protocol or method.

**Reviewer Confidence:**

4: Quite sure. I tried to check the important points carefully. It's unlikely, though conceivable, that I missed something that should affect my ratings.

---

> ### Author Rebuttal · Authors · 2023-08-28
>
> _Question A: Are you able to perform some experiments like what I suggested in reasons to reject? I think some things like--showing how your dataset improves model performance on other DC test sets, how other datasets don't perform as well on your (or their) DC test sets, and how other DC models from existing datasets don't help downstream MT as much as models trained with your dataset do--would greatly strengthen the claim that your dataset is useful._
>
> **Author Response:**
>
> Thank you for your suggestion.
>
> After testing models finetuned on various datasets to analyse the robustness of models using the DISCO dataset, we present results of a few additional experiments below.
>
> We get the following results for English DC -
>
> | Test Dataset | Model name | Test F1 score |
> |----------|----------|----------|
> | DISCO  | MuRIL finetuned on DISCO (English only)  | 96.65 |
> |   | MuRIL finetuned on Switchboard  | 86.97  |
> |  | MuRIL finetuned on LARD  | 89.33  |
> | Switchboard  | MuRIL finetuned on DISCO (English only) | 95.67 |
> | | MuRIL finetuned on Switchboard | 88.44 |
> | | MuRIL finetuned on LARD | 89.33 |
> | LARD  | MuRIL finetuned on DISCO (English only) | 96.21 |
> | | MuRIL finetuned on Switchboard | 91.23 |
> | | MuRIL finetuned on LARD | 97.30 |
>
> Similarly we get the following results for Hindi DC -
>
> | Test Dataset | Model name | Test F1 score |
> |----------|----------|----------|
> | DISCO  | MuRIL finetuned on DISCO (Hindi only) | 94.29 |
> |   | MuRIL finetuned on Kundu et al. (2022)  | 81.16  |
> | Kundu et al. (2022) (only 100 sentences)| MuRIL finetuned on DISCO (Hindi only) | 87.16 |
> |   | MuRIL finetuned on Kundu et al. (2022)  | 84.81 |
>
>
> Thus we observe that in all cases, models trained on the DISCO dataset, perform better than models trained on real or synthetic data. For LARD, our model performs comparatively even in out of domain synthetic test sets. Such experiments could not be performed for French and German because of lack of other open source DC datasets in these languages.
>
> We were also able to compare the performance of other DC systems on the Hindi - English DISCO MT improvement task when used with a state-of-the-art MT system (NLLB). The following BLEU scores were obtained when using the different models as described above:
>
> | Model name | BLEU score |
> |----------|----------|
> | MuRIL finetuned on DISCO (Hindi only)   | 42.63 |
> | MuRIL finetuned on Kundu et al. (2022)  | 38.97  |
>
> These results additionally strengthen the validity of our dataset as a good resource to train DC systems for downstream MT improvement in conversational settings.
>
> **Additional Response:**
>
> We would like to highlight that our proposed dataset can be used to train DC systems that can be used as a preprocessing step before any downstream processing such as machine translation, intent classification, etc. Since existing SOTA MT systems are not trained on disfluent text, such a pre-processing step can improve downstream performance as illustrated in section 5.2. Additionally, the data we created for disfluent Hindi - English and German - English translation tasks can be used to train better performing conversational MT systems.
>
> **Final Comments**
>
> The authors sincerely thank the reviewer for their time to assess and evaluate our submission.
>
> We apologize for the lack of clarification of these above points and would be happy to include these explanations in the revised version of this paper in appropriate tables and sections.

---

### Meta-Review · Area_Chair_d2yW · 2023-09-18

**Recommendation:** 3

**Metareview:**

We thank the authors for their submission.

This paper introduces a new Disfluency Correction (DC) dataset in English, French, German, and Hindi. It then presents many methods (RNN, Transformers, etc.)

Reasons to Accept:
- An practically valuable dataset for a potentially very useful under-studied but useful task (as pointed out by R1)
- Large range of baseline scores reported on the dataset (as pointed out by R1)
- The dataset is further showcasing the importance of DC for Machine Translation

Reasons to Reject:
- A single annotator per language: The paper would benefit from annotator agreement reported at least on a subsample of the data.
- Better acknowledgment and discussion of related work.

Suggestions:
Evaluating the performance of rule-based predictions on DISCO would help us understand how challenging the dataset and task is (e.g., removing repetition with simple pattern matching.)

Typos Grammar Style And Presentation Improvements:
- Most tables do not follow the EMNLP template

---

### Decision · Program_Chairs · 2023-10-07

**Decision:**

Accept-Findings

**Comment:**

We thank the authors for their submission.

This paper introduces a new Disfluency Correction (DC) dataset in English, French, German, and Hindi. It then presents many methods (RNN, Transformers, etc.)

Reasons to Accept:
- An practically valuable dataset for a potentially very useful under-studied but useful task (as pointed out by R1)
- Large range of baseline scores reported on the dataset (as pointed out by R1)
- The dataset is further showcasing the importance of DC for Machine Translation

Reasons to Reject:
- A single annotator per language: The paper would benefit from annotator agreement reported at least on a subsample of the data.
- Better acknowledgment and discussion of related work.

Suggestions:
Evaluating the performance of rule-based predictions on DISCO would help us understand how challenging the dataset and task is (e.g., removing repetition with simple pattern matching.)

Typos Grammar Style And Presentation Improvements:
- Most tables do not follow the EMNLP template